# Low CD8 T Cell Counts Predict Benefit from Hypoxia-Modifying Therapy in Muscle-Invasive Bladder Cancer

**DOI:** 10.3390/cancers15010041

**Published:** 2022-12-21

**Authors:** Vicky Smith, Debayan Mukherjee, Anna Maria Tsakiroglou, Alexander Baker, Hitesh Mistry, Ananya Choudhury, Peter Hoskin, Timothy Illidge, Catharine M. L. West

**Affiliations:** 1Division of Cancer Sciences, University of Manchester, Manchester M13 9PL, UK; 2Cancer Research UK Manchester Institute, Manchester M20 4BX, UK; 3Division of Pharmacy and Optometry, University of Manchester, Manchester M13 9PL, UK; 4Manchester Academic Health Science Centre, Manchester M13 9NQ, UK; 5Christie Hospital NHS Foundation Trust, Manchester M20 4BX, UK; 6Mount Vernon Cancer Centre, Northwood HA6 2RK, UK

**Keywords:** hypoxia, immunotherapy, ICI, biomarker, bladder cancer, TME, tumour microenvironment, multiplex, TILs, CD8 T cells

## Abstract

**Simple Summary:**

Precision medicine is needed for muscle-invasive bladder cancer to improve survival rates due to high rates of tumour recurrence and poor patient outcomes. Hypoxia and an immunosuppressive tumour microenvironment are both independent poor prognostic factors that are targetable by hypoxia-modification and immune checkpoint inhibitors (ICIs), respectively. CD8 T cell counts are showing promise as a predictive biomarker for a durable response to ICIs. However, there is currently no biomarker to stratify patients to either hypoxia-targeting or immune-targeting therapies. Using data from a phase III trial where patients were randomised to radiotherapy +/− hypoxia-modifying therapy, we demonstrate that low CD8 T cell counts predict benefit from hypoxia-modifying therapy and associate with high hypoxia. These results inform the design of a clinical trial using CD8 T cell count as a biomarker for stratification to receive either radiotherapy with hypoxia-modifying therapy (low CD8 counts) or standard-of-care alone or with an ICI (high CD8 counts).

**Abstract:**

Background: As hypoxia can drive an immunosuppressive tumour microenvironment and inhibit CD8+ T cells, we investigated if patients with low tumour CD8+ T cells benefitted from hypoxia-modifying therapy. Methods: BCON was a phase III trial that randomised patients with muscle-invasive bladder cancer (MIBC) to radiotherapy alone or with hypoxia-modifying carbogen plus nicotinamide (CON). Tissue microarrays of diagnostic biopsies from 116 BCON patients were stained using multiplex immunohistochemistry (IHC) with the markers CD8, CD4, FOXP3, CD68 and PD-L1, plus DAPI. Hypoxia was assessed using CA9 IHC (*n* = 111). Linked transcriptomic data (*n* = 80) identified molecular subtype. Relationships with overall survival (OS) were investigated using Cox proportional hazard models. Results: High (upper quartile) vs. low CD8 T cell counts associated with a better OS across the whole cohort at 16 years (n = 116; HR 0.47, 95% CI 0.28–0.78, *p* = 0.003) and also in the radiotherapy alone group (*n* = 61; HR 0.39, 95% CI 0.19–0.76, *p* = 0.005). Patients with low CD8+ T cells benefited from CON (*n* = 87; HR 0.63, 95% CI 0.4–1.0, *p* = 0.05), but those with high CD8 T cells did not (*n* = 27; *p* = 0.95). CA9 positive tumours had fewer CD8+ T cells (*p* = 0.03). Prognostic significance of low CD8+ T cells in the whole cohort remained after adjusting for clinicopathologic variables. Basal vs. luminal subtype had more CD8+ cells (*p* = 0.02) but was not prognostic (*n* = 80; *p* = 0.26). Exploratory analyses with other immune markers did not improve on findings obtained with CD8 counts. Conclusions: MIBC with low CD8+ T cell counts may benefit from hypoxia-modifying treatment.

## 1. Introduction

Bladder cancer is the 10th most commonly diagnosed cancer worldwide. In Europe, ~200,000 people are diagnosed with bladder cancer and ~67,000 deaths are attributed to the disease each year [1]. Stage one bladder cancer is classified as a non-muscle invasive disease, stages two and three are localised muscle-invasive bladder cancer (MIBC), and stage four is metastatic MIBC. In the UK, the five-year survival rate is 52.6% for all stages combined, which decreases from 79.4% for diagnosis at stage one to 45.7% and 41.2% for diagnosis at stages two and three, respectively [2]. National Institute for Health and Care Excellence, United Kingdom guidelines for localised MIBC stipulates a treatment strategy consisting of neoadjuvant chemotherapy followed by radical cystectomy or radiotherapy with a radiosensitiser [3]. Although radiotherapy-based regimens are often reserved for patients considered poor candidates for surgery, meta-analyses showed no difference in survival outcome and fewer complications with radiotherapy [4]. Given the multiple current and emerging options, there is a need to identify biomarkers for personalising treatments to optimise patient outcomes after radiotherapy and improve survival rates [5].

Hypoxia and an immunosuppressive tumour microenvironment (TME) are both poor prognostic factors that contribute to radiotherapy and chemotherapy resistance in MIBC [6,7]. They are targetable by hypoxia-modifying therapy and immune checkpoint inhibitors (ICIs), respectively, and are therefore of interest as potential biomarkers. Adding hypoxia-modifying carbogen and nicotinamide (CON) to radiotherapy improves survival outcomes in bladder cancer patients [8,9]. This benefit is only seen in patients with the most hypoxic tumours [10].

There is growing evidence that hypoxia drives a suppressive immune TME. Specifically, hypoxia decreases CD8 T cell activity and proliferation in the TME [11]. In mouse models, systemic oxygenation and locally delivered oxygen ameliorated this effect, restored the anti-tumour cytotoxic effects of CD8 T cells and improved the efficacy of an ICI [12,13,14,15]. We hypothesised that patients with low tumour CD8 T cell counts might benefit from hypoxia-targeting CON given with radiotherapy. 

There are currently six new ICIs approved for advanced/metastatic bladder cancer. Despite the initial response only around one in five patients have sustained effects, and so there is a need to find biomarkers that predict ICI response [16,17]. A systematic review and meta-analysis of 33 studies involving multiple cancers, including bladder, showed high tumour infiltrating CD8+ T cells associated with benefit from ICIs whether given as single-agents or as part of combination treatments [18]. Therefore, the work presented here is important for the design of a future biomarker-driven trial where MIBC patients with high CD8 T cells could be randomised to receive standard-of-care treatment with or without an ICI; patients with low CD8 T cells would receive radiotherapy plus CON. 

Hypoxia can also drive an immunosuppressive TME by increasing pro-tumour immune infiltrates, such as regulatory T helper cells (Tregs; CD4+FOXP3+) and macrophages (CD68+), and the immunosuppressive immune checkpoint molecule PD-L1 [19]. Our previous research showed more hypoxic MIBC had increased expression of PD-L1 [20]. PD-L1 is also of current interest as a potential biomarker and predictor of response to ICI treatment [21]. 

Therefore, our primary objective in this study was to investigate whether patients with low tumour CD8 T cell counts benefited from having CON with radiotherapy. Our study involved tumour samples collected from patients enrolled in the bladder CON (BCON) trial that randomised MIBC patients to radiotherapy alone or with CON. Secondary objectives were to (1) carry out additional exploratory analyses to investigate other suppressive immune markers (CD4, FOXP3, CD68, PD-L1) in the context of hypoxia and prognosis; and (2) compare findings with molecular subtypes as an emerging biomarker for bladder cancer. 

## 2. Methods

### 2.1. BCON Cohort

BCON was a prospective multicentre phase III clinical trial that randomised patients from 2000–2006 registered as CRUK/01/003. The trial protocol is described in detail elsewhere [9]. Tissue microarrays (TMAs) were previously made from diagnostic biopsies of muscle-invasive tumour samples as described elsewhere [22]. The updated long-term clinical outcomes were used in all analyses [8].

### 2.2. Multiplex Staining Protocol 

Multiplex immunohistochemistry (IHC) was performed using the automated Ventana system (Ventana Medical Systems, Oro Valley, AZ, USA). The Opal^TM^ detection system (Akoya Biosciences, Marlborough, MA, USA) was used as it allows for repeated cycles of staining and stripping with antibodies of the same species without cross-reactivity, which enables simultaneous staining of five targets in the same tissue section. The staining protocol involved initial deparaffinisation and epitope retrieval at pH 8.5 followed by multiple cycles of incubation with primary antibody, secondary antibody and Opal^TM^ detection. The cycles were separated by a short denaturation at pH 6. UltraMap anti-rabbit/mouse HRP conjugated secondary antibodies were used (Roche, Basel, Switzerland). Antibody and Opal^TM^ concentrations are listed in Appendix A. After the automated protocol the slides were manually washed by three 5 min cycles of a 1:10 dilution of EZ preparation (Ventana Medical Systems) before being counter-stained with a 1:120,000 dilution of DAPI (ThermoFisher, MA, USA) for five minutes. The slides were then cover-slipped with ProLong^TM^ Gold Antifade Mountant (ThermoFisher). 

### 2.3. Multispectral Scanning and Unmixing

A Vectra 3 microscope (Akoya Biosciences) was used to scan the slides with a Vectra Fluorescence Illuminator 200 Watt Metal Halide Bulb set to 10%. Manual annotation of TMA core locations was performed using a low resolution scan at 4× magnification. A multi-spectral image of each core was then acquired at 20× magnification using DAPI, AF488, TRITC, AF594 and AF647 filters. After multispectral scanning, the slides were spectrally unmixed using inForm software 2.4.9 (Akoya Biosciences) and a pre-prepared spectral library. The library was prepared using single-plex controls to acquire the individual spectrum of each fluorophore, alongside DAPI and autofluorescence, under the same experimental parameters as the multiplexed slides. 

### 2.4. Data Analysis

Image analysis was performed in HALO 3.2 with the TMA Module (Indica Labs, Albuquerque, NM, USA) to manually exclude artefacts and then obtain counts of each marker per core. A representative core is shown in Appendix A. The marker quantification was then exported to R where normalisation was performed by calculating the percentage of each marker from the total cell count across all cores of each patient (*n* = 116). This method was chosen to account for cores of different sizes and different numbers of cores per patient. CA9 IHC scores previously generated were used to assign patients a CA9 status of present or absent (*n* = 111) [22]. CA9 status was used instead of our hypoxia signature, as using the hypoxia signature was not significant due to reduced patient numbers. Gene expression data previously generated [10] was used to assign each patient a molecular subtype using the “consensusMIBC” package in R [23]. 

### 2.5. Statistical Methods

All the analyses were performed using R version 4.0.5 and RStudio version 1.3.1093 and associated packages used to analyse data and calculate significance. Non-parametric statistics (Wilcox test) were used due to the non-normal distribution of each marker. Relationships with overall and local progression-free survival for 16-year follow up were assessed in R using Cox proportional hazard models and Kaplan–Meier curves via the “survival” and “survminer” packages which uses a log-rank test to calculate *p* values. Tables of characteristics were tabulated using “table1” package in R. Cumulative incidence curves were plotted and the significance calculated using “cmprsk” package. Multivariable analysis and statistical tests were performed using the “survival” package and the results were tabulated using “gtsummary” package.

## 3. Results

### 3.1. Study Cohort

The study cohort comprised 333 patients and TMAs with 353 cores. Following staining and filtering out poor quality cores due to folds/dropouts/scanning errors, cores (*n* = 301) from 116 patients were available for analysis (Appendix A). Each patient had 1–3 different 1 mm cores analysed, with an average of 2 cores per patient. Appendix A shows no differences in the clinicopathologic characteristics of the study cohort compared with the BCON trial cohort. 

### 3.2. Tumours with Low Tumour CD8+ Cells Associate with a Poor Prognosis 

The percentage of CD8+ cells ranged from 0 to 43.52% per patient (Appendix A). There was a non-linear relationship between the percentage of CD8+ cells and overall survival. CD8+ T cell counts stratified into quartiles showed a significant but non-linear relationship with overall survival (*p* = 0.014) with the upper quartile an outlier (Appendix A). The upper quartile (9.52–43.52%) was used to stratify patients into CD8 high and low groups. Patients with CD8-high tumours had significantly better overall (HR 0.47, 95% CI 0.28–0.78, *p* = 0.003) and local progression free (HR 0.52, 95% CI 0.32–0.87, *p* = 0.011) survival (Figure 1). 

### 3.3. Low Tumour CD8+ Cell Counts Predict Benefit from Hypoxia Modification

High CD8+ cells were a good prognostic factor in patients who received radiotherapy alone for overall (*n* = 61; HR 0.39, 95% CI 0.19–0.76, *p* = 0.005) and local progression free (HR 0.38, 95% CI 0.19–0.75, *p* = 0.004) survival (Figure 2A,C). High CD8+ counts were not prognostic in patients who had radiotherapy plus CON for both overall (*p* = 0.17) and local progression free (*p* = 0.55) survival (Figure 2B,D). Patients with low CD8+ counts had better overall survival when CON was given with radiotherapy (HR 0.63, 95% CI 0.4–1.0, *p* = 0.05) and a trend towards improved local progression free survival (HR 0.63, 95% CI 0.4–1.0, *p* = 0.052) (Figure 3A,C). Those with high CD8 counts derived no benefit from hypoxia modification (Figure 3B,D). Appendix A shows similar clinicopathologic characteristics for patients with low versus high tumour CD8 counts. 

### 3.4. Low CD8+T Cell Counts Associate with CA9 Positivity but Retain Independent Prognostic Significance

CA9 IHC staining previously performed on the samples and used as a marker of hy-poxia shows positive versus negative tumours had fewer CD8+ cells (*p* = 0.03, *n* = 111; Figure 4A). In order to investigate further the relationship between hypoxia and CD8+ T cell counts, the patients were stratified into four groups according to both CA9 status and CD8+ counts. Cumulative incidence curves were generated, instead of survival curves due to subsequent low event numbers per group, and showed that regardless of the hypoxia status, both groups with high CD8 had fewer events for both overall (*p* = 0.011) and local progression free (*p* = 0.029) survival compared to groups with low CD8 counts (Figure 4B,C). Multivariable Cox proportional hazard model analyses for overall survival, taking into account other clinicopathologic factors, showed the percentage of CD8+ T cells main-tained prognostic significance (HR 0.33 95% CI 0.19–0.60; *p* < 0.001) alongside CA9 (*p* = 0.045), necrosis (*p* = 0.004), treatment arm (*p* = 0.031), and age (*p* = <0.001; Table 1).

### 3.5. Tumours with Low CD8+ Counts Are More Likely to Have a Luminal Molecular Subtype

Percentage of tumour CD8+ cells were higher in the basal/squamous and stroma-rich subtypes (Figure 5A). We further stratified the subtypes into first order subtyping to either luminal or basal and excluded the five NE-like patients. The percentage of CD8+ cells were significantly higher in tumours with a basal versus a luminal molecular subtype (*p* = 0.02; Figure 5B). Kaplan–Meier curves using the first order subtyping showed no prognostic significance for survival between the two subtypes in our cohort (*n* = 80; *p* = 0.26; Figure 5C). 

### 3.6. Exploratory Analyses with PD-L1, Macrophages and Other T Cell Types

Of the other markers, only CD4 T helper cells counts were positively associated with CA9 expression (*p* = 0.007; Appendix A). However, there was no significant association with overall survival (*p* = 0.36). None of the other markers showed any significant prognostic relevance. Appendix A summarises the range of counts obtained with the other multiplexed immune markers studied. There was significantly higher expression of CD68 and PDL1 in the basal group compared to the luminal group (*p* = 0.037 and *p* < 0.001, respectively; Appendix A). 

## 4. Discussion

Our study found low tumour CD8+ T cell counts are associated with a poor overall prognosis, in keeping with previously published findings in urothelial cancers [21,24]. We used whole TMA cores with no tumour/stroma differentiation. Some authors reported that tissue compartmentalisation is important when using CD8 T cells as a prognostic marker. However, a recent meta-analysis found that, although there is a slight superiority to quantifying stromal T cells, CD8+ T cell levels are prognostic regardless of localisation pattern [18]. Deng et al. also demonstrated the prognostic significance of CD8+ T cells on formalin-fixed, paraffin-embedded (FFPE) whole tumour sections in bladder cancer [25]. Here, we demonstrate for the first time that patients with low CD8+ T cells benefit from hypoxia modifying therapy, CON. Another novel finding from this study is that bladder tumours with low CD8+ T cell counts have higher CA9 expression (marker of tumour hypoxia), but that CD8+ T cell count retains independent prognostic significance in multivariable analysis. 

Hypoxia is an adverse prognostic factor in bladder cancer. We showed previously that the presence of necrosis associates with hypoxia and is a negative prognostic factor in the BCON cohort and that patients with tumour necrosis benefit from CON [8,22]. Similarly, patients with a high expression of the hypoxia markers HIF1a and CA9 had significantly improved 5-year local relapse free survival with RT+CON compared to RT alone [26]. Our group subsequently developed a 24-gene hypoxia-associated signature for bladder cancer. A high hypoxia score was a negative prognostic factor and predicted benefit of therapeutic intervention with CON [10]. 

The results presented here show that assessing CD8+ T cell status provides additional information to assessing tumour hypoxia. While we identify a group of patients who benefit from having CON with radiotherapy, the literature indicates that a high CD8+ T cell count predicts benefit from ICIs in both bladder and other cancer types [18,27]. Therefore, using CD8 as a biomarker could stratify patients to receive radiotherapy plus CON (low CD8+ tumours) or standard-of-care treatment with/without an ICI (high CD8+ tumours). A recent paper showed MIBC patients with low stromal TILs (which correlated with CD8+ levels) have a poor outcome following radical cystectomy compared to those with higher levels of stromal TILs, suggesting those with low TILs would benefit from alternative treatment options [28]. Further research is needed to investigate the prognostic significance of CD8 counts in MIBC patients undergoing radiotherapy with chemotherapy (another standard-of-care treatment), but we would hypothesise that low CD8 counts would also be an adverse prognostic factor.

CD8+ T cells have a crucial role in the effectiveness of radiotherapy to elicit anti-tumour immune responses after treatment [29]. Radiotherapy, both directly and indirectly, causes immunogenic cell death of the tumour cells, which relies upon sufficient recruitment and activation of antigen-specific effector CD8+ T cells to elicit an effective anti-tumour immune response [30]. A recent study analyzing the gene expression of MIBC tumours found that higher expression of genes reflecting CD8+ T cell infiltration associated with improved survival for patients receiving radiotherapy-based treatment, but not those receiving chemotherapy-based treatment [31]. This further highlights the usefulness of considering the immune contexture in MIBC to predict radiotherapy responses and the key role of CD8+ T cells in radiotherapy-induced, immune-mediated killing.

In this study, we found that MIBC with low CD8+ count had worse survival after radiotherapy-based treatment, and were more hypoxic, than those with high CD8+ count. Hypoxia might reduce CD8+ T cell infiltration via a mechanism involving increased adenosine signalling. Adenosine accumulation occurs in tissue as hypoxia increases and this has a strong role in the regulation of tumour inflammation. Specifically, extracellular adenosine has been linked with decreasing T cell differentiation and activity in hypoxic environments. It has been shown that T cells either avoid going into, or are inhibited by, hypoxic and adenosine rich areas due to the adenosine receptor signalling pathway [13]. Adenosine has been shown to bind to adenosine receptors on effector T cells, reducing activity and differentiation [32]. Further to this, hypoxia has been shown to induce T cell apoptosis by the adenosine receptor signalling pathway [33]. 

Importantly, we found that CON improves outcomes in low CD8+ MIBC. Increasing oxygenation with CON might reduce adenosine signalling and promote T cell infiltration to increase anti-tumour immunity. It has previously been shown in a murine model that hyperoxic breathing (60% oxygen) decreased levels of HIF1a and extracellular adenosine. The hyperoxic breathing was also shown to upregulate MHC class I on the tumours’ cells, which is known to enable CD8 T cell anti-tumour effects [12]. A recent study demonstrated that locally delivered oxygen reduced adenosine accumulation in hypoxic cells and restored the cytotoxic ability of CD8 T cells both in vitro and in vivo [14]. A further recent study in a murine model demonstrated that hyperbaric oxygen directly increased T cell infiltration [15]. Both carbogen and nicotinamide could be recapitulating these effects to restore the anti-tumour capacity of CD8 T cells in the tumour. As molecular subtype is increasingly of interest as a biomarker in bladder cancer, we subtyped our dataset using the consensusMIBC classification and discovered a higher proportion of most infiltrates in the basal/squamous and stroma-rich subtypes, as reported by others [23]. However, molecular subtype was not prognostic in our cohort. Molecular subtypes were previously examined on a larger BCON sub-cohort with available transcriptomic data, where tumours were stratified into basal and luminal subtypes using a different methodology, and it was also found not to be prognostic [8]. In this study we also investigated other suppressive immune markers including PD-L1, CD68+ macrophage populations and Treg populations characterized by CD4+ and FOXP3+ co-localisation. In agreement with the published literature we found PDL1 had no clinical relevance as a biomarker in this context [34]. None of the other markers showed any prognostic relevance or link with CA9. This was unexpected as it has been shown in other cancer types and in vivo models that hypoxia drives an increase in both Tregs and macrophages [11,35]. 

The limitations of our study include the relatively low numbers of samples and limited tissue available for analysis. Further work is needed to elucidate mechanisms of hypoxia causing low CD8+ T cells. Analysis needs to be done on a bladder cancer ICI clinical trial cohort with transcriptomics and available FFPE blocks. It could then be determined if those tumours with low CD8+ T cells are more hypoxic and if those patients with high CD8 T cells benefitted from the ICI. Tumour/stroma stratification could be evaluated to see if prognostic strength can be further improved. However, current analysis shows prognostic significance is independent of tissue differentiation, which would make any future test simpler and cheaper as it would allow use of the tissue and stain indiscriminately, reducing costs and turnaround time. 

## 5. Conclusions

Bladder cancer remains an area of unmet clinical need with high rates of tumour recurrence; alongside more effective treatments options, patient stratification for personalised treatment is needed to further improve survival rates. Importantly for developing novel MIBC treatment approaches, hypoxia and immunosuppressive TME are independent negative prognostic factors that can both be therapeutically targeted. Our results are hypothesis generating and inform the design of clinical trials where patients are stratified using CD8+ T cell count as a biomarker to receive either RT+CON or other oxygen modifying approaches (in patient tumours with low CD8+ counts) or randomised to standard-of-care alone or with an ICI (high CD8+ counts). 

## Figures and Tables

**Figure 1 cancers-15-00041-f001:**
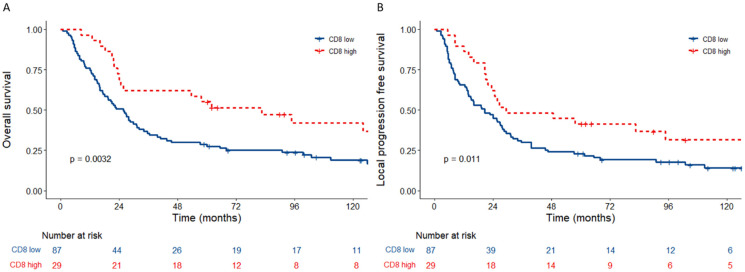
Kaplan–Meier plots of survival according to percent CD8. *p* values are from log-rank tests calculated by R packages “survival” and “survminer”. CD8 T cells are stratified into low and high using the upper quartile as the cut-off. (**A**) Overall survival and (**B**) local progression free survival in the whole cohort according to percent CD8 T cells low vs. percent CD8 T cells high for 16-year follow-up data.

**Figure 2 cancers-15-00041-f002:**
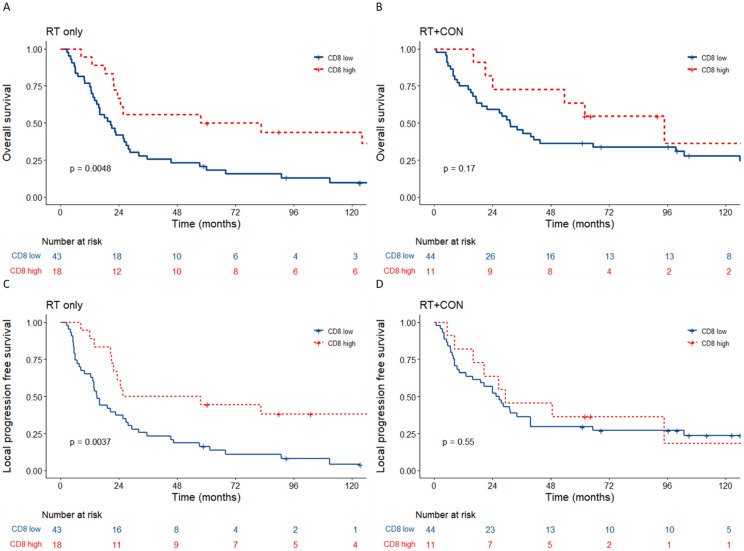
Kaplan–Meier plots for the treatment arms of BCON according to percent CD8 T cells. *p* values are log-rank tests calculated as previously described. Overall survival for (**A**) RT only treatment arm and (**B**) RT+CON treatment arm according to percent CD8 low vs. percent CD8 high. Local progression free survival for (**C**) RT only treatment arm and (**D**) RT+CON treatment arm according to percent CD8 low vs. percent CD8 high.

**Figure 3 cancers-15-00041-f003:**
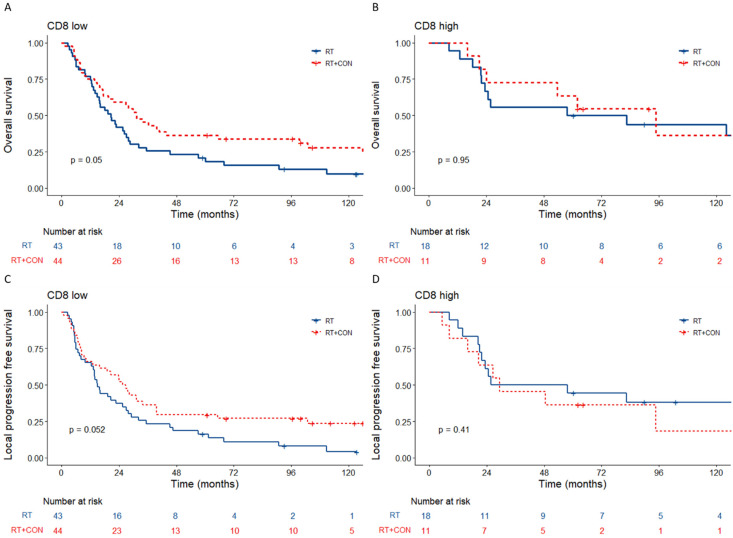
Kaplan–Meier plots for CD8 low and high according to treatment arm of BCON. *p* values are log-rank tests calculated as previously described. Overall survival for (**A**) CD8 low and (**B**) CD8 high according to RT only vs. RT+CON treatment arms. Local progression free survival for (**C**) CD8 low and (**D**) CD8 high according to RT only vs. RT+CON treatment arms.

**Figure 4 cancers-15-00041-f004:**
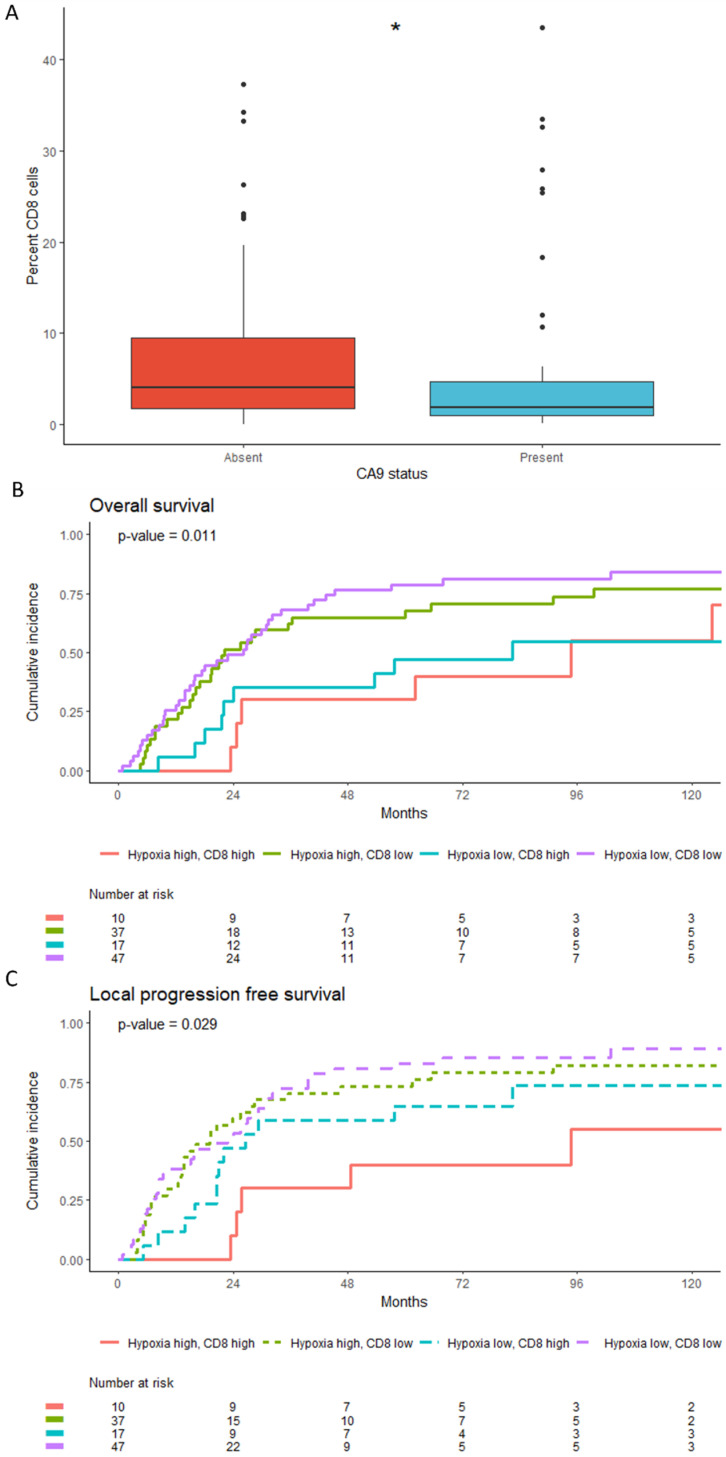
(**A**) Boxplot showing the population density for CD8 when grouped into CA9 absent or present. *p* values < 0.05 are represented by *. (**B**) Cumulative incidence of events for overall survival and (**C**) local progression free survival according to percent CD8 low/high, hypoxia low/high. Tumours were stratified into hypoxia low or high using the CA9 IHC status and CD8 high or low using the upper quartile as done previously. *p* values are calculated using the “cmprsk” package in R.

**Figure 5 cancers-15-00041-f005:**
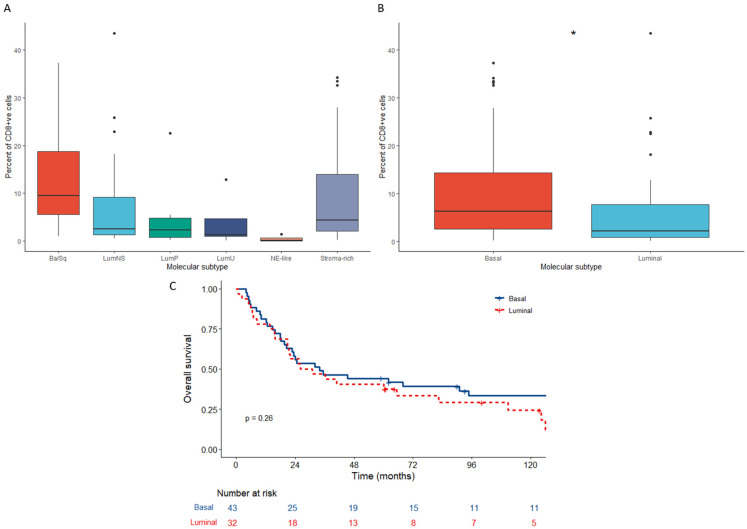
Boxplots showing the association between the percent of CD8 and the molecular subtype as assigned (**A**) using “consensusMIBC” package in R (*n* = 80) and (**B**) further stratification into basal (basal/squamous and stroma-rich) and luminal (LumP, LumU, LumNS) subtypes with NE-like excluded (*n* = 75). Statistical tests are Wilcox tests with *p* values < 0.05 are represented by *. (**C**) Kaplan–Meier plot for overall survival according to first order molecular subtypes (basal/luminal with NE-like excluded; *n* = 75). *p* values are log rank tests calculated using “survival” and “survminer” packages in R.

**Table 1 cancers-15-00041-t001:** Multivariable Cox proportional hazard model analyses of proportion of CD8 T cells alongside other clinicopathologic variables as calculated by “survival” and “gtsummary” packages in R.

Characteristic	HR ^1^	95% CI ^1^	*p*-Value
Age	1.05	1.02, 1.08	<0.001
Gender			
Male	-	-	
Female	1.24	0.65, 2.36	0.51
Tumour stage			
2	-	-	
3, 4	1.03	0.61, 1.76	0.90
Grade			
2	-	-	
3	0.79	0.41, 1.52	0.48
Treatment			
RT	-	-	
RT+CON	0.60	0.38, 0.95	0.031
Tumour de-bulking			
Complete	-	-	
Partial	0.92	0.54, 1.56	0.76
Biopsy	1.73	0.95, 3.16	0.075
CA9			
Absent	-	-	
Present	0.60	0.37, 0.99	0.045
Necrosis			
Absent	-	-	
Present	2.12	1.27, 3.52	0.004
Percent of CD8 cells			
Low	-	-	
High	0.33	0.19, 0.60	<0.001

^1^ HR = Hazard Ratio, CI = Confidence Interval.

## Data Availability

The data underlying this article will be shared on reasonable request to the corresponding author.

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
