# Peer review of "Low CD8 T Cell Counts Predict Benefit from Hypoxia-Modifying Therapy in Muscle-Invasive Bladder Cancer"

_cancers, 2022, doi:10.3390/cancers15010041_

Round 1
Reviewer 1 Report
The authors did a beautiful work investigating if patients with low tumor CD8+ T-cells benefitted from hypoxia-modifying therapy.
The contents are attractive and feasible in clinical practice.
However, the results showed in this study are ambiguous to make the final conclusion. Please see the following comments.
Results:
line190-191) The statistical values were not suitable to state as this sentence, however, the survival plots seemed tendency.
line 194) Table S4: Did each clinicopathologic factors in each group show significant differences? Please show the p-value.
line 213-215): 1) The figure showed low CD8 had better OS, please check. 2) Did authors compare the high and low CD8 group without division of high, low hypoxia? The result that "regardless of the hypoxia status, both groups with high CD8 had significantly fewer events for local progression-free survival compared to those groups with low CD8 counts" is ambiguous in Fig.4C.
line 215-220) It is better to conduct the Cox regression analysis before the survival analysis to rule out the insignificant factors first. In your results, the CA9 did not have an independent significance that needed not to include in the survival analysis then.
Reviewer 2 Report
The Manuscript "Low CD8 T cell counts predict benefit from hypoxia-modifying therapy in muscle-invasive bladder cancer" Can be accepted after minor revisions
Comments
-Extensive scientific English editing is required
-This paper discusses the role of Hypoxia and CD8 infiltration in muscle-invasive bladder cancer. It's still not clear to readers about the staining hypoxia and CD8 infiltration. Authors must include and explain the staining in the main figures. (CA9 staining must be included)
-
Author Response
Point: Extensive scientific English editing is required
Response: The manuscript has been written by native English speakers and reviewed by two other reviewers who do not think it requires any English editing. Based on this we do not think the manuscript requires further English editing. If reviewer 3 insists then please could some examples be provided where they believe extensive English editing is required?
Point: This paper discusses the role of Hypoxia and CD8 infiltration in muscle-invasive bladder cancer. It's still not clear to readers about the staining hypoxia and CD8 infiltration. Authors must include and explain the staining in the main figures. (CA9 staining must be included)
Response: The multiplexed IHC staining is extensively described and explained in the manuscript methods section under the subheading Multiplex Staining Protocol and a representative core is shown in Supplementary Figure 1, as referenced on line 144 and shown in Supplementary Materials. The CA9 staining and scoring was performed previously by others within the lab group as described in the methods section under the subheading Data Analysis with the relevant paper referenced (Ref 22). The referenced paper extensively details the CA9 staining and scoring methodology.
Reviewer 3 Report
The authors described the relationship between the benefit from hypoxia-modifying treatment and CD8+ T-cell counts. As we know, the response of ICIs is about 20%, effective prognostic markers are needed clinically. The finding is important however, some significant revisions are needed to be accepted by the journal.
1.some details should be improved.
Line 69, Line77 there are two abbreviations for TME, Line 77 “tumour microenvironment” can be deleted.
Pigure 4, Line 223," ns = not significant, *<0.05, **<0.01, ***<0.001." I did not see the P <0.01 or <0.001, the "**<0.01, ***<0.001." can be deleted. The same condition is in the annotation of Figure 5 (Line 245).
2.The possible mechanism (not only tumour hypoxia) of CD8+ T cell affecting the prognosis of bladder cancer can be discussed in the discussion part in combination with some literature.
Round 2
Reviewer 1 Report
The revised version is acceptable.
Author Response
Thank you.